# Characterization of full-length *CNBP* expanded alleles in myotonic dystrophy type 2 patients by Cas9-mediated enrichment and nanopore sequencing

**Massimiliano Alfano[1†], Luca De Antoni[1†], Federica Centofanti[2], Virginia Veronica Visconti[2], Simone Maestri[1], Chiara Degli Esposti[1], Roberto Massa[3], Maria Rosaria D'Apice[4], Giuseppe Novelli[2,5,6], Massimo Delledonne[1,7], Annalisa Botta[2*], Marzia Rossato[1,7*]**

[1]Department of Biotechnology, University of Verona, Verona, Italy; [2]Department of Biomedicine and Prevention, Medical Genetics Section, University of Rome Tor Vergata, Rome, Italy; [3]Department of Systems Medicine (Neurology), University of Rome Tor Vergata, Rome, Italy; [4]Laboratory of Medical Genetics, Tor Vergata Hospital, Rome, Italy; [5]IRCCS Neuromed, Via Atinense, Molise, Italy; [6]Department of Pharmacology, School of Medicine, University of Nevada Reno, Reno, United States; [7]Genartis s.r.l., Via P. Mascagni, Castel D'azzano, Italy

**\*For correspondence:**
botta@med.uniroma2.it (AB);
marzia.rossato@univr.it (MR)

[†]These authors contributed
equally to this work

**Abstract** Myotonic dystrophy type 2 (DM2) is caused by CCTG repeat expansions in the *CNBP* gene, comprising 75 to >11,000 units and featuring extensive mosaicism, making it challenging to sequence fully expanded alleles. To overcome these limitations, we used PCR-free Cas9-mediated nanopore sequencing to characterize *CNBP* repeat expansions at the single-nucleotide level in nine DM2 patients. The length of normal and expanded alleles can be assessed precisely using this strategy, agreeing with traditional methods, and revealing the degree of mosaicism. We also sequenced an entire ~50 kbp expansion, which has not been achieved previously for DM2 or any other repeat-expansion disorders. Our approach precisely counted the repeats and identified the repeat pattern for both short interrupted and uninterrupted alleles. Interestingly, in the expanded alleles, only two DM2 samples featured the expected pure CCTG repeat pattern, while the other seven presented also TCTG blocks at the 3′ end, which have not been reported before in DM2 patients, but confirmed hereby with orthogonal methods. The demonstrated approach simultaneously determines repeat length, structure/motif, and the extent of somatic mosaicism, promising to improve the molecular diagnosis of DM2 and achieve more accurate genotype–phenotype correlations for the better stratification of DM2 patients in clinical trials.

## Editor's evaluation

The study utilized a long-read sequencing to correctly assess expanded alleles in DM2 patients and has technical and diagnostic value. We anticipate more valuable approaches like this in the near future.

## Introduction

Myotonic dystrophy type 2 (DM2; MIM#602668) is an autosomal dominant multisystem disorder characterized by progressive proximal muscle weakness, myotonia, myalgia, calf hypertrophy, and

multiorgan involvement with cataract, cardiac conduction defects, and endocrine disorders (*Meola and Cardani, 2015*; *Montagnese et al., 2017*). The disease is caused by a $(CCTG)_n$ repeat expansion in intron 1 of the *CNBP* gene (previously *ZNF9*; MIM*116955) on chromosome 3q21.3 (*Liquori et al., 2001*). The CCTG repeat tract is part of a complex $(TG)_v (TCTG)_w (CCTG)_x$ motif that, in healthy-range alleles, is generally interrupted by one or more GCTG, TCTG, or ACTG (NCTG) motifs that confer repeat stability (*Radvanszky et al., 2013*; *Mahyera et al., 2018*; *Guo and Lam, 2016*). Nonpathogenic alleles contain up to 26 CCTG repeat units, whereas premutations are composed of <75 'pure' CCTG blocks whose clinical significance remains unclear (*Mahyera et al., 2018*; *Botta et al., 2021*). In DM2 patients, the number of repeats is between ~75 and >11,000 units, among the largest reported so far in repeat-expansion disorders (*Depienne and Mandel, 2021*). The DM2 mutation shows marked somatic instability and tends to increase in length over time within the same individual, but it does not show a strong bias toward intergenerational expansion, and genetic anticipation is rarely seen in DM2 families (*Liquori et al., 2001*; *Kamsteeg et al., 2012*; *Mahyera et al., 2018*; *Thornton, 2014*; *Udd et al., 2003*).

The few genotype–phenotype studies reported so far in DM2 patients did not reveal any significant associations between the severity of the disease, including the age at onset, and the number of CCTG repeats (*Day et al., 2003*; *Udd et al., 2003*). The identification of such correlations is hindered by heterogeneity across tissues, somatic instability, and the technical challenges that must be overcome to measure repeat lengths accurately in such large microsatellite expansions. This has prevented the discovery of additional *in cis* genetic modifiers that may ameliorate or exacerbate DM2 disease symptoms. The genetic features of the *CNBP* microsatellite locus, as well as its extreme length and high CG content, have frustrated attempts to size and sequence expanded alleles in DM2 patients. Even the investigators in the original gene-discovery study were unable to sequence the entire CCTG array because of its length and the high level of somatic mosaicism (*Liquori et al., 2001*; *Bachinski et al., 2003*; *Day et al., 2003*).

According to international guidelines, current best practice for DM2 genetic testing relies mainly on PCR-based approaches (*Botta et al., 2006*; *Kamsteeg et al., 2012*). These include an initial short-range PCR (SR-PCR) to exclude a DM2 diagnosis when two normal-range alleles are detected. When only one allele is visible, $(CCTG)_n$ expansions can be identified by long-range PCR (LR-PCR) or quadruplet-repeat primed PCR (QP-PCR), leading to a ~99% detection rate (*Kamsteeg et al., 2012*). However, neither method can define the exact length of large DM2 expansions, which requires the Southern blot analysis of digested genomic DNA and has a sensitivity of ~80% (*Day et al., 2003*). The latter method is time consuming, requires large amounts of DNA, and is not included in the routine workflow of most diagnostic centers. Moreover, none of the methods described above can resolve the expansion to the single-nucleotide level and they have a limited ability to detect minor alleles and the degree of somatic mosaicism.

Third-generation long-read sequencing technologies provide an unprecedented opportunity to fully characterize DM2 expansions in terms of repeat size, allele configuration, and base composition. Whereas most second-generation sequencing methods produce short reads, third-generation methods such as Oxford Nanopore Technologies (ONT) and PacBio SMRT sequencing facilitate the analysis of DNA fragments multiple kilobases in length, including large repetitive elements and their flanking regions. Furthermore, third-generation methods are generally based on PCR-free workflows, thus avoiding amplification-related biases (*Hommelsheim et al., 2014*) and challenges caused by regions with a high CG content (such as the *CNBP* locus). However, these technologies are more expensive and error-prone than first- and second-generation methods. Targeted approaches that maximize data production on a selected region of interest can compensate for such errors and provide cost-effective alternatives to whole-genome sequencing.

Targeted enrichment approaches coupled to long-read sequencing have already been used for the in-depth characterization of repeat expansions in fragile X syndrome (*FMR1*), Huntington's disease (*HTT*), and neuronal intranuclear inclusion disease (*NOTCH2NLC*), although these expansions are significantly shorter than the $(CCTG)_n$ repeats in DM2 patients (*DeJesus-Hernandez et al., 2021*; *Ebbert et al., 2018*; *Giesselmann et al., 2019*; *Grosso et al., 2021*; *Hafford-Tear et al., 2019*; *Höijer et al., 2018*; *Mizuguchi et al., 2021*; *Sone et al., 2019*; *Tsai et al., 2017*; *Wallace et al., 2021*; *Wieben et al., 2019*; *Mitsuhashi and Matsumoto, 2020*). The longest allele thus far characterized by third-generation sequencing is a 21-kbp allele of the *C9orf72* gene that causes amyotrophic lateral

sclerosis and frontotemporal dementia (*DeJesus-Hernandez et al., 2021*). In addition, many of the works cited above combined PacBio SMRT sequencing with LR-PCR (*Mangin et al., 2021*; *Ciosi et al., 2021*; *Cumming et al., 2018*), which is unsuitable for the analysis of DM2 expansions due to their extreme length and high GC content. Importantly, PacBio reads do not exceed 20 kbp in a PCR-free enrichment (*DeJesus-Hernandez et al., 2021*; *Ebbert et al., 2018*; *Hafford-Tear et al., 2019*; *Höijer et al., 2018*; *Tsai et al., 2017*; *Wieben et al., 2019*), and would not completely span DM2 pathogenic expansions (20 kbp on average, but up to 50 kbp).

To address these issues, we assessed the analysis of *CNBP* expansions using a combination of CRISPR/Cas9-based enrichment (Cas9-enrichment) and ONT sequencing. The latter can generate reads >100 kbp in length (*Payne et al., 2019*; *Iyer et al., 2022*) and recently demonstrated valuable for the analysis of very long repetitive elements, like telomeric and centromeric regions in the completion of human genome (*Sergey et al., 2022*; *Consortium and The Telomere-to-telomere T T, 2022*) and microsatellite expansions in the pathogenic range (*Stevanovski et al., 2022*). In this manner, we sequenced full-length $(CCTG)_n$ expansions in nine DM2 patients including one mutated allele 47 kbp in length. Because this approach achieves single-nucleotide resolution, we were able to detect a previously unreported $(TCTG)_n$ motif at the 3' end of the *CNBP* expansion in seven of the DM2 patients. Our pilot study demonstrated that Cas9-mediated enrichment and long-read sequencing improves the DM2 diagnostic workflow, facilitating the in-depth characterization of *CNBP* expansions by accurately reporting the repeat length, structure/motif, and degree of somatic mosaicism in a single analysis. In the future, this approach may enable more precise genotype–phenotype correlations and thus improve patient stratifications in clinical trials for personalized therapies.

## Results

### Molecular characterization of DM2 patients using traditional methods

We analyzed nine DM2 patients (six males and three females, mean age = 46.4 ± 20 years) with existing molecular diagnoses based on a combination of PCR-based approaches (SR-PCR, LR-PCR, and QP-PCR) to detect the presence of $(CCTG)_n$ expansions in the *CNBP* gene (*Table 1*). Four patients were familial cases (A1–A4) (*Figure 1—figure supplement 1*) whereas patients B, C, D, E, and F were sporadic cases. The maximum length of the $(CCTG)_n$ expansion could not be determined using routine diagnostic methods, so we digested genomic DNA and estimated the size of each allele by Southern blot analysis (*Figure 1A*). This suggested that the size of the microsatellite ranged from 20 to about 40 kbp (*Figure 1—figure supplement 2* and *Table 1*). As expected, no signal was detected for healthy control subjects (CTR) or myotonic dystrophy type 1 (DM1) patient (*Figure 1—figure supplement 2*). The characterization of normal *CNBP* alleles by SR-PCR and Sanger sequencing revealed the presence of eight short interrupted alleles with the structure $(TG)_{17-24}$ $(TCTG)_{6-9}$ $(CCTG)_{5-7}$ GCTG CCTG TCTG $(CCTG)_7$ and one short uninterrupted allele with the structure $(TG)_{19}$ $(TCTG)_9$ $(CCTG)_{12}$, matching our previous results for an Italian population (*Botta et al., 2021*; *Table 1*).

### *CNBP* repeat-expansion analysis by Cas9-mediated enrichment coupled to ONT sequencing

We characterized the full-length *CNBP* expansions at single-nucleotide resolution by ONT sequencing following Cas9-mediated enrichment. Accordingly, we designed two gRNAs to excise a 4.2-kbp fragment spanning the *CNBP* repeat on chromosome 3q21.3 (*Figure 1A* and *Supplementary file 1*).

Genomic DNA from the nine DM2 patients was analyzed in four singleplex and four multiplex runs, the latter applied to clinical samples here for the first time (*Supplementary file 1*). Cas9-mediated sequencing achieved good target coverage in all experiments, with 346 ± 64 reads (mean ± standard error of the mean) on the *CNBP* locus (*Figure 1B* and *Supplementary file 1*). Singleplex runs had consistently lower background than multiplex runs (0.08× vs. 0.57×), and thus achieved a higher average fold enrichment (3521- vs. 637-fold) (*Figure 1B, C* and *Supplementary file 1*). Collectively, for each DM2 patient, we generated a mean total of 105,737 PASS reads, 308 of which were on target (*Figure 1D*) and 186 of which completely spanned the normal or expanded alleles (*Figure 1E*). Only these 'complete sequences' were used for subsequent analysis, representing ~78% and ~22% of the normal and expanded repeat-spanning reads, respectively (*Figure 1E*).

**Table 1.** Demographic and molecular features of the dystrophy type 2 (DM2) patients.
For each patient, the table shows the ID, sex, age, age at onset, repeat length, and structure of the *CNBP* normal allele characterized by Sanger sequencing, and the estimated repeat length of the expanded allele determined by Southern blotting (–, not available).

| Sample ID | Sex | Age | Age at onset | Normal allele | | Expanded allele |
| --- | --- | --- | --- | --- | --- | --- |
| | | | | Repeat length (bp) | Repeat structure | Repeat length (bp) |
| A1 | F | 75 | 70 | 136 | $(TG)_{24}$ $(TCTG)_7$ $(CCTG)_5$ GCTG CCTG TCTG $(CCTG)_7$ | 40,000 |
| A2 | M | 27 | 25 | 130 | $(TG)_{17}$ $(TCTG)_9$ $(CCTG)_5$ GCTG CCTG TCTG $(CCTG)_7$ | 20,000 |
| A3 | M | 21 | – | 132 | $(TG)_{20}$ $(TCTG)_8$ $(CCTG)_5$ GCTG CCTG TCTG $(CCTG)_7$ | 20,323 |
| A4 | M | 65 | 61 | 122 | $(TG)_{19}$ $(TCTG)_9$ $(CCTG)_{12}$ | 32,745 |
| B | F | 49 | 44 | 134 | $(TG)_{21}$ $(TCTG)_7$ $(CCTG)_6$ GCTG CCTG TCTG $(CCTG)_7$ | 40,000 |
| C | M | 20 | – | 140 | $(TG)_{24}$ $(TCTG)_6$ $(CCTG)_7$ GCTG CCTG TCTG $(CCTG)_7$ | 29,027 |
| D | M | 44 | 39 | 134 | $(TG)_{19}$ $(TCTG)_9$ $(CCTG)_5$ GCTG CCTG TCTG $(CCTG)_7$ | 20,000 |
| E | F | 61 | 43 | 134 | $(TG)_{19}$ $(TCTG)_9$ $(CCTG)_5$ GCTG CCTG TCTG $(CCTG)_7$ | 30,000 |
| F | M | 56 | 50 | 138 | $(TG)_{21}$ $(TCTG)_9$ $(CCTG)_5$ GCTG CCTG TCTG $(CCTG)_7$ | 39,000 |

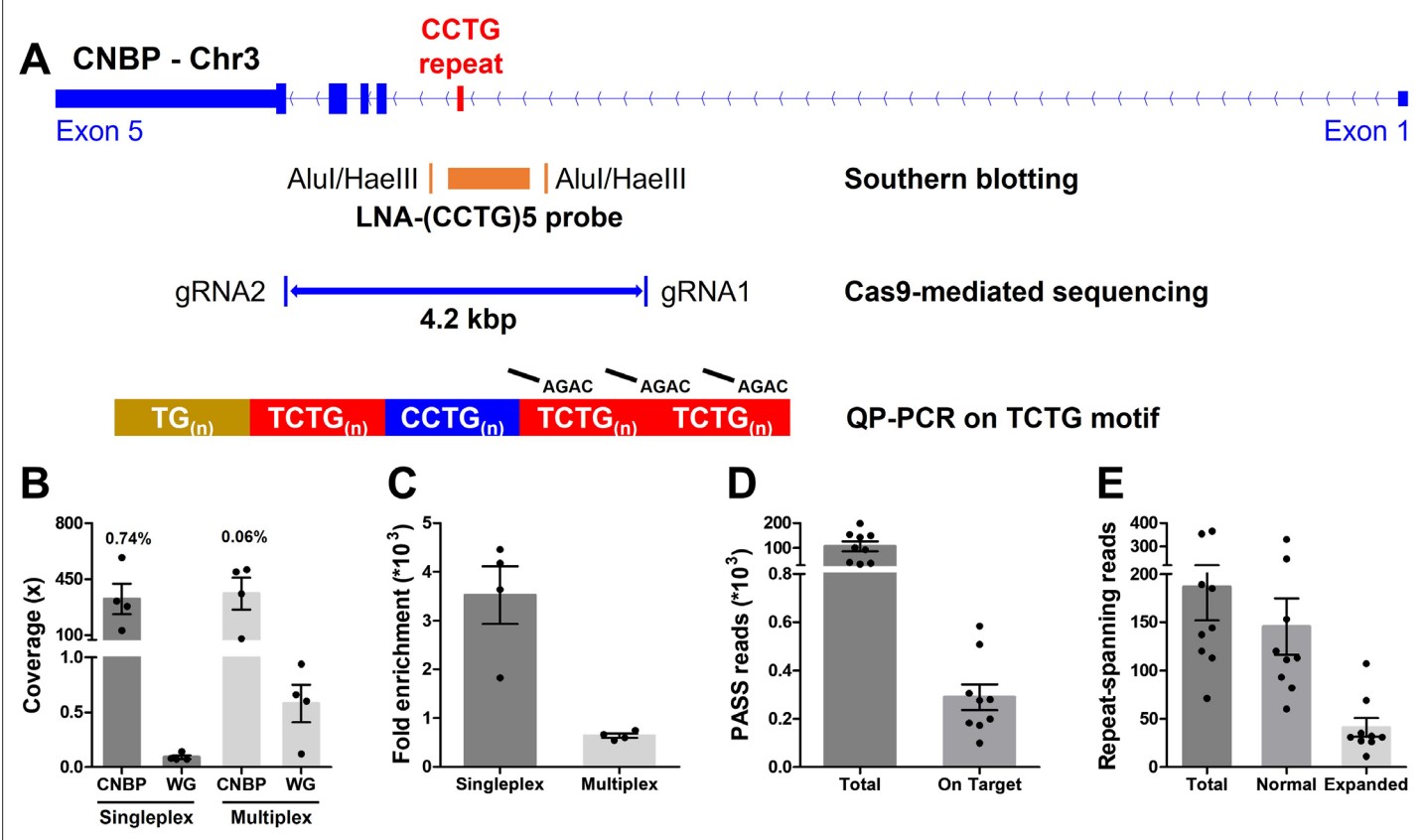

**Figure 1.** Cas9-mediated sequencing of the *CNBP* microsatellite. (**A**) Experimental methods applied retrospectively to study the *CNBP* microsatellite in nine confirmed dystrophy type 2 (DM2) patients. The positions of AluI and HaeIII restriction sites (142 bp upstream and 108 bp downstream of the CCTG array, respectively) and the (CCTG)$_5$ probe (orange rectangle) for Southern blot hybridization are shown, along with the gRNAs cleavage site for Cas9-mediated enrichment (boundaries of blue line = 4.2 kb) and the annealing position of P4TCTG primers for quadruplet-repeat primed PCR (QP-PCR). (**B**) Average coverage and (**C**) fold enrichment obtained by Cas9-mediated enrichment in DM2 patients following singleplex (*n* = 4) or multiplex (*n* = 4) runs. Numbers above bars represent the percentage of on-target reads. (**D**) Total and on-target number of PASS reads generated for each DM2 patient and (**E**) number of reads fully spanning the *CNBP* microsatellite and attributable to either the normal or expanded alleles. Data are plotted as means ± standard error of the mean (SEM). WG, whole genome. Sequencing statistics of singleplex and multiplex experiments are reported in detail in *Supplementary file 1*.

The online version of this article includes the following source data and figure supplement(s) for figure 1:

**Figure supplement 1.** Pedigree of the dystrophy type 2 (DM2) family analyzed.

**Figure supplement 2.** Southern blot analysis of expanded alleles in dystrophy type 2 (DM2) patients.

**Figure supplement 2—source data 1.** Original scan of the Southern blot analysis reported in *Figure 1—figure supplement 2*.

**Figure supplement 2—source data 2.** *Figure 1—figure supplement 2* including the original uncropped scan of the Southern blot analysis.

The de novo assembly of reads derived from the normal *CNBP* alleles in DM2 patients (145 on average per sample, IQR = 67; *Figure 1E*) showed that the complex (TG)$_v$ (TCTG)$_w$ (CCTG)$_x$ (NCTG)$_y$ (CCTG)$_z$ repeat ranged in size from 122 to 141 bp, corresponding to 12–15 CCTG quadruplets (*Table 2* and *Figure 2B*). The size and repeat pattern in each patient were largely consistent with the Sanger sequencing data (99.5% mean accuracy, Pearson's *r*=0.971, p < 0.0001; *Figure 2C*), with six patients showing a perfect match, two differing at a single-nucleotide position and only one differing at two nucleotide positions (*Table 2*).

Reads derived from the expanded alleles (41 on average per sample, IQR = 11; *Figure 1E*) ranged from 344 bp to as much as 46.6 kbp (*Figure 2D*), confirming the presence of extremely large expansions in these patients. To our knowledge, the latter is the longest repeat expansion analyzed thus far at single-nucleotide resolution (*Mizuguchi et al., 2021*; *Sone et al., 2019*; *Giesselmann et al., 2019*; *Wallace et al., 2021*) and is one of the longest DNA fragments captured by Cas9-mediated enrichment with no specific adjustment (*Gilpatrick et al., 2020*; *Iyer et al., 2020*). Considering average

**Table 2.** *CNBP* repeat analysis based on Cas9-mediated sequencing of the normal alleles.
For each patient, the table shows the characteristics of normal *CNBP* alleles based on the analysis of ONT sequencing data, in terms of length and structure.
The table reports the percentage identity of consensus sequences reconstructed from ONT or Sanger sequencing data and the incongruences in ONT-derived sequences are highlighted in bold.

| Normal allele | Sample ID | Repeat length (bp) | Repeat structure | Identity with Sanger sequence |
|---|---|---|---|---|
| | A1 | 136 | $(TG)_{24}$ $(TCTG)_7$ $(CCTG)_5$ GCTG CCTG TCTG $(CCTG)_7$ | 100.0% |
| | A2 | 131 | $(TG)_{17}$ **TG**CTG $(TCTG)_8$ $(CCTG)_5$ GCTG CCTG TCTG $(CCTG)_7$ | 99.2% |
| | A3 | 132 | $(TG)_{20}$ $(TCTG)_8$ $(CCTG)_5$ GCTG CCTG TCTG $(CCTG)_7$ | 100.0% |
| | A4 | 122 | $(TG)_{19}$ $(TCTG)_9$ $(CCTG)_{12}$ | 100.0% |
| | B | 134 | $(TG)_{21}$ $(TCTG)_7$ $(CCTG)_6$ GCTG CCTG TCTG $(CCTG)_7$ | 100.0% |
| | C | 141 | $(TG)_{24}$ **TG**CTG $(TCTG)_5$ $(CCTG)_7$ GCTG CCTG TCTG $(CCTG)_7$ | 99.3% |
| | D | 138 | $\mathbf{(TG)_{21}}$ $(TCTG)_9$ $(CCTG)_5$ GCTG CCTG TCTG $(CCTG)_7$ | 97.1% |
| | E | 134 | $(TG)_{19}$ $(TCTG)_9$ $(CCTG)_5$ GCTG CCTG TCTG $(CCTG)_7$ | 100.0% |
| | F | 138 | $(TG)_{21}$ $(TCTG)_9$ $(CCTG)_5$ GCTG CCTG TCTG $(CCTG)_7$ | 100.0% |

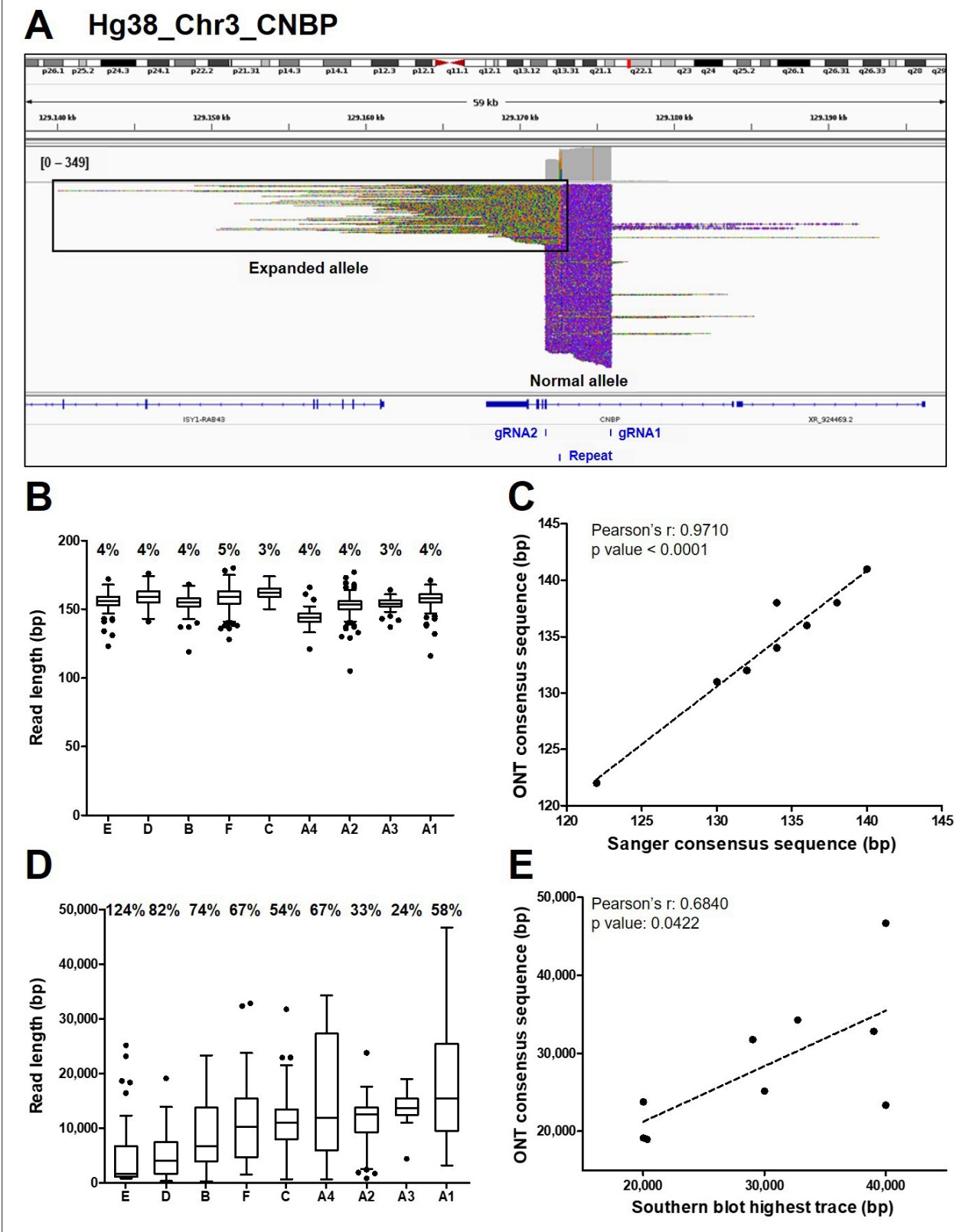

**Figure 2.** Analysis of ONT sequencing data from normal and expanded *CNBP* alleles. (**A**) Integrative Genomics Viewer (IGV) visualization of ONT sequencing data at the *CNBP* locus of a representative dystrophy type 2 (DM2) patient following Cas9-mediated enrichment. Reads generated from the normal allele feature clear cuts on both sides of the *CNBP* repeat, whereas those derived from the expanded allele are longer, soft-clipped and do not match the reference genome, as expected. Length distributions of reads derived from the normal alleles (**B**) and expanded alleles (**D**) of each

*Figure 2 continued on next page*

*Figure 2 continued*

patient. Boxes represent the interquartile range (IQR) of lengths, the horizontal line is the median, whiskers and outliers are plotted according to Tukey's method. (**C**) Correlation between the length of ONT and Sanger consensus sequences for the normal allele (n = 9). (**E**) Correlations between the maximum length of ONT sequences (longest complete read) and the upper edge of the Southern blot trace for the expanded allele (n = 9). Numbers on top of panels (**B**) and (**D**) indicate the coefficient of variation of normal and expanded alleles, respectively.

values per sample, the number of repetitive quadruplets varied from 1371 to 4421, corresponding to expansion lengths of 5485–17,685 bp. Moreover, in each patient, the longest molecule sequenced with ONT (derived from the expanded allele) largely agreed with the Southern blot estimates (Pearson's r = 0.6840, p = 0.0422, *Figure 2E*), with the exception of sample (B). Even within the same individual, the mean size of reads derived from the expanded allele was consistently more variable than that from the normal allele (65% ± 0.091 vs. 4% ± 0.002; *Figure 2B–D*, *Figure 3*). Such pronounced variability within each DM2 patient indicates extensive mosaicism, in agreement with previous reports (*Liquori et al., 2001*; *Bachinski et al., 2003*; *Day et al., 2003*).

To characterize the repeat pattern across the expanded microsatellite locus, we identified the quadruplet motifs in each individual read, and highlighted them with distinct colors after aligning 'complete sequences' at the 5' and 3' ends (*Figure 3* and *Figure 3—figure supplement 1A, B*). The total number of quadruplets was highly variable within each patient, thus confirming the extensive donor-dependent mosaicism described above (*Figure 3C* and *Figure 3—figure supplement 1*). For example, reads from patient C carried on average 3000 quadruplets (*Figure 3C*), but the number of (CCTG) repeats in individual reads varied from 150 up to 8000 (*Table 3*). Since we analyzed only those

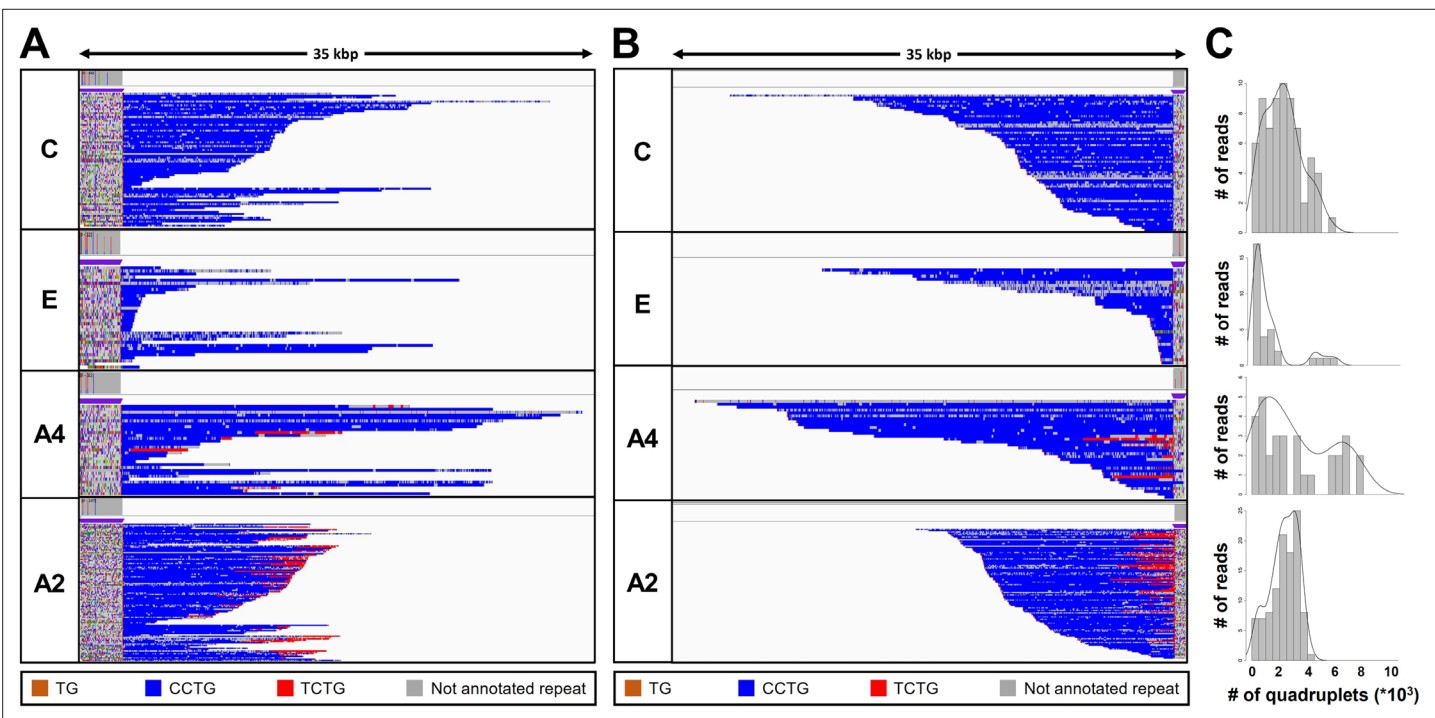

**Figure 3.** Analysis of the expanded-repeat *CNBP* alleles in dystrophy type 2 (DM2) patients. Integrative Genomics Viewer (IGV) visualization (35-kbp windows) of ONT-targeted sequencing data from the expanded alleles of four representative DM2 patients. Complete reads were aligned at the 5' end (**A**) and then at the 3' end (**B**) in order to identify the repeat pattern that characterizes the expanded microsatellite locus. Each motif in the expanded alleles was visualized using a different color, as indicated in the key. Samples C and E contained a 'pure' CCTG expansion (blue) whereas samples A4 and A2 also contained the unexpected TCTG motif (red) downstream of CCTG. (**C**) Abundance of quadruplets identified in each patient. The *y*-axis shows the number of ONT reads with a certain number of repeats, whereas the *x*-axis shows the number of quadruplet repeats identified. ONT reads were grouped into 500 bp bins. The gray line represents the estimated kernel density of the underlying solid gray distribution of ONT reads.

The online version of this article includes the following figure supplement(s) for figure 3:

**Figure supplement 1.** Analysis of the *CNBP* repeat motif for the expanded alleles in dystrophy type 2 (DM2) patients.

**Figure supplement 2.** Analysis of the *CNBP* 5'-end (TG)$_v$ repeat motif of the CNBP expanded alleles of A1–A4 dystrophy type 2 (DM2) family members.

**Table 3.** *CNBP* repeat analysis based on Cas9-mediated sequencing of the expanded alleles.

For each patient, the table shows the characteristics of expanded *CNBP* alleles based on the analysis of ONT sequencing data, in terms of length and structure. The expanded $(CCTG)_x$ are colored in blue and $(TCTG)_y$, in red. Potential incongruences in ONT-derived sequences are highlighted in bold. The table indicates also the fraction of reads carrying the unexpected TCTG motif at the 3' end of the *CNBP* microsatellite expansion.

| Expanded allele | Sample ID | Repeat length (bp) (*min–max*) | Repeat structure | Number of reads carrying the TCTG motif |
|---|---|---|---|---|
| | A1 | 3241–46,685 | $\mathbf{(TG)_{20}}(TCTG)_7(CCTG)_{1000-12,000}(TCTG)_{0-10}$ | 14 (45%) |
| | A2 | 864–23,779 | $(TG)_{18}(TCTG)_7(CCTG)_{1000-4500}(TCTG)_{0-2000}$ | 92 (86%) |
| | A3 | 4429–18,983 | $\mathbf{(TG)_{19}}(TCTG)_7(CCTG)_{3000-5000}(TCTG)_{0-25}$ | 8 (73%) |
| | A4 | 660–34,284 | $(TG)_{18}(TCTG)_7(CCTG)_{250-8000}(TCTG)_{0-1500}$ | 9 (29%) |
| | B | 344–23,358 | $(TG)_{18}(TCTG)_7(CCTG)_{300-4000}(TCTG)_{0-400}$ | 6 (23%) |
| | C | 700–31,753 | $(TG)_{20}(TCTG)_7(CCTG)_{150-8000}$ | 0 (0%) |
| | D | 383–19,143 | $(TG)_{18}(TCTG)_7(CCTG)_{100-4000}(TCTG)_{0-1000}$ | 3 (11%) |
| | E | 848–25,162 | $(TG)_{18}(TCTG)_6(CCTG)_{200-6200}$ | 0 (0%) |
| | F | 1533–32,824 | $(TG)_{15}(TCTG)_{10}(CCTG)_{400-6000}(TCTG)_{0-2000}$ | 15 (43%) |

reads containing at least 600 bp up-/downstream the *CNBP* microsatellite, and thus comprising the repeat entirely, we excluded that such large variability could be ascribed to the analysis of fragmented DNA molecules.

A variable number of $(TG)_v$ repetitions upstream of the $(CCTG)_n$ array was also observed in the familiar cases A1–A4 (*Table 3*). Since this microsatellite tract is supposed to be stably transmitted within the same family, the observed discrepancy was likely an artefact due to ONT accuracy, as reported for the healthy alleles. Manual inspection of sequencing data indeed confirmed that all family members show an equivalent pattern of $(TG)_v$ repetitions (*Figure 3—figure supplement 2*).

The uninterrupted $(TG)_v$ $(TCTG)_w$ $(CCTG)_x$ motif that characterizes expanded *CNBP* alleles was found at the 5' end of the repeat locus in all nine patients (*Figure 3A*, *Figure 3—figure supplement 1*, and *Table 3*). However, only two patients featured a 'pure' pattern of $(CCTG)_n$ repeats. In the remaining seven, we observed additional $(TCTG)_n$ arrays (colored in red) at the 3' end of the CCTG expansion, which has been never reported in DM2 patients before (*Figure 3B*, *Figure 3—figure supplement 1*, and *Table 3*). When present, the TCTG motif was detected in a highly variable fraction of sequences (11%–86% of the expanded allele reads, *Table 3*), and differed widely in length (40–8000 bp) between donors and within the same individual (*Figure 3*, *Figure 3—figure supplement 1*, and *Table 3*).

## Analysis of the TCTG repeat using orthogonal methods

To confirm the presence of the $(TCTG)_n$ motifs in the *CNBP* expanded alleles, we used a traditional QP-PCR method for the selective amplification of TCTG blocks at the 3' end of the $(CCTG)_n$ array (*Figure 1A*). In agreement with the ONT data, QP-PCR analysis using primer P4TCTG revealed an electrophoretic profile compatible with the presence of $(TCTG)_n$ downstream of the $(CCTG)_n$ expansion in seven DM2 samples (*Figure 4A*). The intensity and pattern of fluorescent peaks obtained using primer P4TCTG were more variable across samples compared to the routine protocol using the standard P4CCTG primer, possibly due to different levels of somatic mosaicism in the TCTG and CCTG expansions. A 260 bp signal was visible in all samples, including the DM2 negative controls (one DM1 positive patient and one healthy subject CTR), suggesting it was a PCR artifact (*Figure 4*). Interestingly, the two patients with 'pure' $(CCTG)_n$ expansions based on ONT data (patients C and E) did not yield amplification peaks within the expansion range with the P4TCTG primer (*Figure 4B*). Fluorescent signals <140 bp were visible in these samples because normal *CNBP* alleles also contain $(TCTG)_n$ repeats at their 5' end (*Table 1*). The direct sequencing of QP-PCR products generated using primer P4TCTG confirmed the presence of $(TCTG)_n$ motif in DM2 patients A1, A2, A3, A4, B, D, and F, thus supporting the ONT data (*Figure 4* and *Figure 4—figure supplement 1*).

## Discussion

The analysis of extremely long microsatellite expansions is challenging, preventing the in-depth characterization of *CNBP* mutations underlying DM2 and its relationship with the clinical phenotype. To date, the genotype–phenotype correlation issue in DM2 is still unsolved and relies on a single study from *Day et al., 2003* in which Southern blot analysis was used to determine the length of DM2 mutation. Because of the extremely large size of the CCTG expansions and somatic instability of the repeat, Southern blot fails to detect the DM2 mutation in about 20% of known carriers, whose expansion length remains undeterminable. Moreover, currently no diagnostic method allows to sequence through fully expanded *CNBP* microsatellites. Here, we have demonstrated for the first time the use of Cas9-mediated enrichment and ONT long-read sequencing to analyze the full $(CCTG)_n$ expansion in DM2 patients at single-nucleotide resolution. We were able to characterize normal and expanded *CNBP* alleles, simultaneously revealing the repeat length, structure/motif, and extent of somatic mosaicism, which is not possible with traditional methods, even if several are used in combination.

The paucity of genotype–phenotype data for DM2 reflects the historical inability to determine the size of CCTG repeats, especially in the largest expansions, which traditionally was based on the Southern blot analysis of genomic DNA. This labor-intensive and time-consuming technique is becoming obsolete because it requires large amounts of high-molecular-weight DNA. LR-PCR can be used instead, but the performance of this technique is poor in regions with a high CG content, and it cannot accommodate the very large expanded alleles (>15 kbp) often found in DM2. Therefore, although LR-PCR achieves the sensitive detection of DM2 mutations, the full length of the $(CCTG)_n$

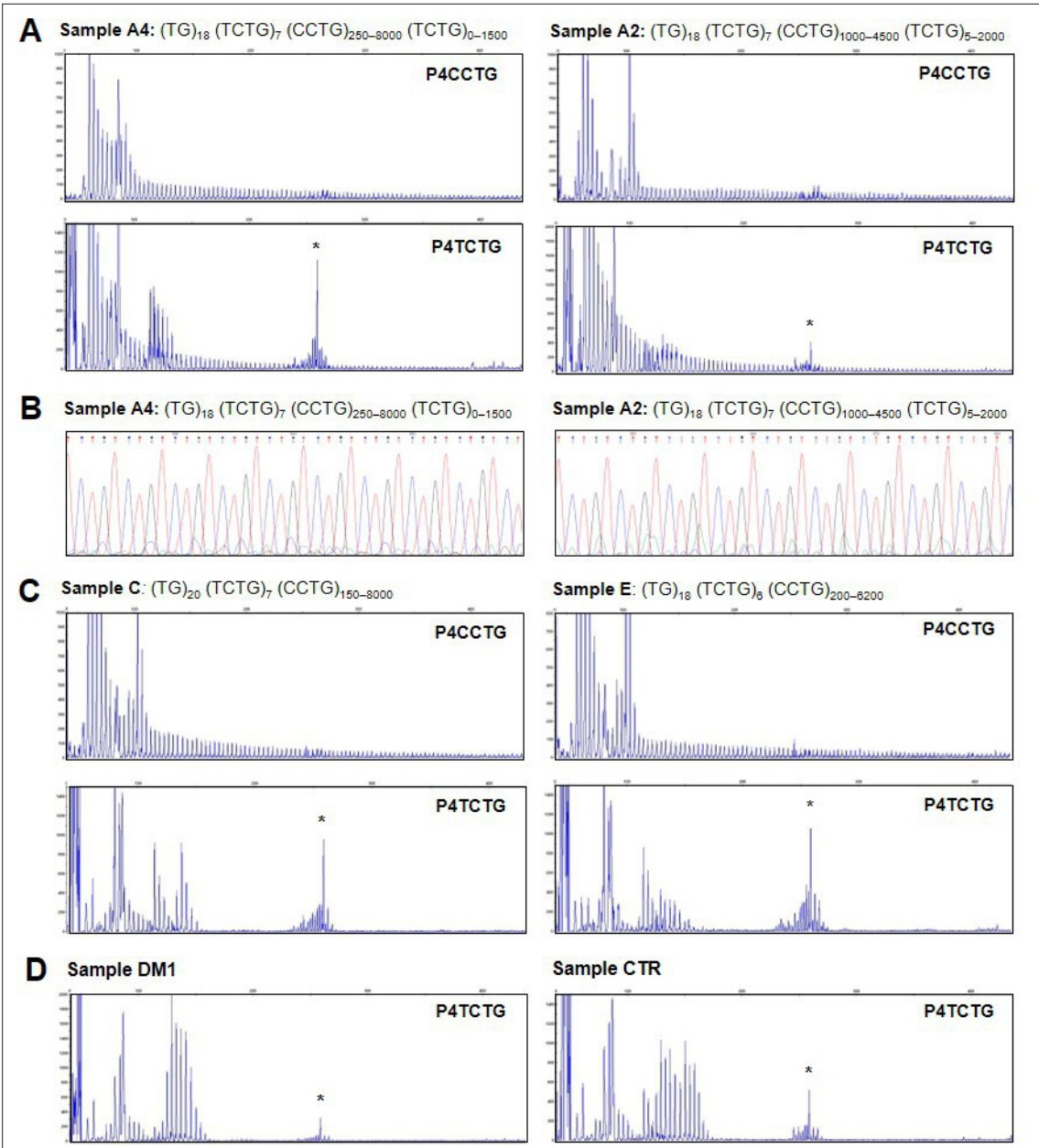

**Figure 4.** Analysis of the TCTG motif by quadruplet-repeat primed PCR (QP-PCR) and Sanger sequencing. (**A**) Representative QP-PCR profiles of genomic DNA samples from patients A2 and A4 and showing the presence of the TCTG block. Upper panels show QP-PCR results using the conventional P4CCTG primer. Lower panels show QP-PCR results using primer P4TCTG. (**B**) Sanger sequencing of the QP-PCR products from P4TCTG reaction confirming the presence of the TCTG sequence. (**C**) QP-PCR profiles of genomic DNA samples from patients C and E showing only the traditional dystrophy type 2 (DM2) motif CCTG. For each patient, the composition of *CNBP* expanded alleles above the QP-PCR tracks reflects the ONT sequencing data. Asterisks (*) indicate nonspecific signals, also visible in the QP-PCR profiles of DM1 and CTR samples (**D**).

The online version of this article includes the following figure supplement(s) for figure 4:

**Figure supplement 1.** Sanger sequencing of quadruplet-repeat primed PCR (QP-PCR) products showing the presence of the $(TCTG)_n$ motif.

expansion cannot be determined in all patients. Our Cas9-targeted sequencing protocol overcomes these limits by focusing long-read sequencing data on the *CNBP* microsatellite, with reads spanning the entire expansion. The length of normal and expanded alleles in DM2 patients determined using this new approach closely matched the values obtained with the traditional reference methods. The mean size of normal *CNBP* alleles was 134 bp whereas expanded alleles ranged from 344 to 46,685 bp, in agreement with previous reports (*Meola and Cardani, 2015*; *Montagnese et al., 2017*;

*Botta et al., 2021*). A single incongruence was observed for one expanded allele in patient B, where ONT sequencing underestimated the size determined by Southern blotting (~20 vs. ~40 kbp). A possible explanation is the presence of damaged DNA in the sample, reflecting its long-term storage in a biobank. Southern blotting involves the fractionation of double-stranded DNA, which would be unaffected by the presence of nicked strands, whereas ONT sequencing involves the analysis of single DNA strands, so the presence of nicks would have a profound effect (*Oxford Nanopore Community, 2021*). Even so, Cas9-mediated enrichment allowed us to sequence DM2 alleles up to ~50 kbp in length at single-nucleotide resolution, which has not been reported previously for DM2 or any other repeat-expansion disorder using this approach (*Giesselmann et al., 2019*; *Mizuguchi et al., 2021*; *Sone et al., 2019*; *Wallace et al., 2021*; *Gilpatrick et al., 2020*; *Iyer et al., 2020*). The analysis of such long and repetitive alleles required the coupling of a PCR-free enrichment protocol to ONT sequencing because even other long-read sequencing technologies cannot accommodate this read length in targeted sequencing experiments. For example, PacBio long-read sequencing was previously used to sequence repeat expansions in DM1, which is also characterized by long alleles of 4–6 kbp, but the microsatellites were first amplified by PCR (*Mangin et al., 2021*; *Cumming et al., 2018*). Even when coupled to PCR-free enrichment approach based on Cas9, the length of PacBio sequencing reads could not exceed 20 kbp (*DeJesus-Hernandez et al., 2021*; *Ebbert et al., 2018*; *Hafford-Tear et al., 2019*; *Höijer et al., 2018*; *Tsai et al., 2017*; *Wieben et al., 2019*).

Although ONT sequencing had been already utilized for the analysis of the microsatellite within the *CNBP* gene, this was confined to CNBP alleles in the normal range only (*Stevanovski et al., 2022*; *Mohammad et al., 2022*). Moreover, the work of Mitsuhashi et al. exploited ONT whole-genome sequencing, that is not applicable in the routine due to the very high costs (*Mohammad et al., 2022*). The group of Stevanovski utilized the recently introduced 'Read Until' feature of ONT sequencing for the analysis of microsatellites in 37 disease-associated loci. This allows the selective sequencing of predefined genomic regions, thus enabling a targeted sequencing with similar advantages of the Cas9-mediated sequencing presented hereby. However, enrichment levels achieved by 'Read Until' (5×) are consistently lower than those obtained with the Cas9 approach (500×), due to higher background (*Stevanovski et al., 2022*). This may constitute an important issue when dealing with extremely long CNBP alleles that can be disadvantaged in sequencing as compared to shorter contaminating fragments (*Holgersen et al., 2021*).

From a technical perspective, we achieved >500× enrichment on *CNBP* using the Cas9 protocol, which is robust and comparable to similar assays for the assessment of microsatellite length (*Giesselmann et al., 2019*; *Mizuguchi et al., 2021*; *Sone et al., 2019*; *Wallace et al., 2021*). We also compared Cas9-mediated singleplex versus multiplex enrichment protocols for the first time on clinical samples and observed a consistently lower performance (10-fold lower enrichment, with 70% unclassified reads) in the multiplex environment, as reported by other ONT users for other type of samples (*Oxford Nanopore Community, 2022*). Further improvements are therefore required before the multiplexing protocol is suitable, for example by combining the Cas9 protocol with a second enrichment step using the 'Read Until' feature of ONT sequencing to exclude the background noise. Alternatively, costs could be optimized by analyzing single samples using Flongles, which produce less sequencing data than regular ONT flow cells but reduce costs by 90%.

The bioinformatic pipeline we used to analyze the *CNBP* microsatellite sequences also allowed us to recognize the repeat pattern and precisely count the repeats, including the short interrupted and uninterrupted alleles typifying the Italian population (*Botta et al., 2021*). Consensus sequences generated from the normal allele shared a high degree of identity (99.5%) and significant correlation with state-of-the-art Sanger sequences, aside from a few nucleotide positions. Similar small discrepancies have also been reported in characterization of the $(TG)_v$ motif upstream of the $(CCTG)_n$ repeated array in familial cases A1–A4. These inconsistences probably reflect ONT sequencing errors and could be addressed by using the most recent base-calling algorithm and eventually the more accurate Q20+ chemistry.

In the expanded alleles of seven DM2 patients, the anticipated $(CCTG)_n$ repeat was accompanied by a previously unreported $(TCTG)_n$ repeat located at the 3' end of the $(CCTG)_n$ array. When present, the atypical motif varied in length between donors and within each sample (40–8000 bp). Repeat interruptions within the expanded array have been reported in 3%–8% of DM1 patients (*Tomé et al., 2018*; *Braida et al., 2010*; *Santoro et al., 2017*; *Santoro et al., 2013*; *Ballester-Lopez et al., 2020*;

*Pešović et al., 2018*; *Miller et al., 2020*; *Radvansky et al., 2011*; *Siena et al., 2018*; *Botta et al., 2017*; *Santoro et al., 2015*; *Addis et al., 2012*; *Fontana et al., 2020*; *Lian et al., 2016*; *Leeflang and Arnheim, 1995*; *Musova et al., 2009*; *Cumming et al., 2018*), but have not been described in DM2 before. This may reflect the challenge of sequencing complete *CNBP* expanded alleles and/or the use of a primer containing 'pure' (CCTG)$_n$ repeats for diagnostic QP-PCR. Given that (TCTG)$_n$ repeats were present in a highly variable proportion of expanded alleles (11%–86%), always in the presence of the typical (CCTG)$_n$ repeats, technical bias may have revealed only the 'pure' (CCTG)$_n$ repeats. Indeed, a modified QP-PCR protocol using a primer containing five TCTG units – (TCTG)$_5$T – was able to confirm the ONT sequencing data. From a biological perspective, the (TCTG)$_n$ motif may have arisen through DNA duplication/repair errors or spontaneous DNA damage in the somatic cells of DM2 patients. Although the presence of this motif may be biologically relevant in the context of DM2, our data must be interpreted with caution. First, the motif was discovered in a small set of patients, most belonging to the same family, so confirmation requires a larger prospective DM2 cohort enrolled in multicenter studies, in which DNA samples are collected in order to ensure the optimal quality for ONT sequencing. Second, considering the known limitations of ONT sequencing when presented with low-complexity regions such as homopolymers, the length and recurrence of such motifs should be investigated using other long-read methods when they are sufficiently advanced to sequence the expanded alleles completely.

The Cas9-targeted sequencing approach also allowed us to estimate the degree of somatic mosaicism for the mutated alleles, either 'pure' or 'interrupted'. On average, the allele length within each patient varied by 65%. Mosaicism plays an important role in the development of disease symptoms, so establishing the relative proportion of expanded alleles in the lower and upper mutation range could add prognostic value, significantly improving genetic counseling for DM2. Extreme mosaicism (more than expected based on previous studies) has also been detected when long-read sequencing is applied to other repeat-expansion disorders, suggesting that such techniques achieve higher resolution (*Loomis et al., 2013*; *Mizuguchi et al., 2021*; *Mangin et al., 2021*). As already demonstrated for DM1 (*Cumming et al., 2019*; *Monckton et al., 1995*), the progenitor allele length (i.e., the length of the CCTG repeat transmitted by the affected parent) is one genetic determinant that influences the age at onset of DM2 symptoms, and that age is further modified by individual-specific differences in the level of somatic instability. Notably, our method accurately distinguished between the shortest expanded allele and the normal allele.

Another advantage of the approach demonstrated hereby is that PCR-free analysis potentially allows the direct assessment of DNA methylation, as already reported for other repeat-linked diseases (*Fukuda et al., 2021*; *Giesselmann et al., 2019*). This can provide additional information concerning the impact of expansions on the functionality of the *CNBP* gene. The methylation of the *CNBP* gene has been analyzed using a pyrosequencing method, revealing hypomethylation of CpG sites upstream and hypermethylation of CpG sites downstream of the (CCTG)$_n$ expansion in DM2 patients and healthy individuals, with no significant differences between these groups (*Santoro et al., 2018*). However, it remains possible that the DM2 mutation could have epigenetic effects in other regulatory regions of the *CNBP* gene and/or in different tissues.

Given the ability of our method to simultaneously determine the size, single-nucleotide composition and degree of somatic mosaicism of DM2 repeat expansions, ONT sequencing could be included in the DM2 diagnostic workflow to improve the information content available for genetic counseling. To date, the cost for Cas9-mediated sequencing of a single patient is relatively high and not comparable with the PCR-based approaches used in the routine of the DM2 molecular diagnostics. Nevertheless, targeted long-read sequencing might help to solve unusual large and complex (CCTG)$_n$ expansions not detectable with conventional methods and identifies noncanonical repetitive motif conformations and sequence interruptions. Taken together, this information will allow more precise correlation between the length and composition of DM2 expansion and the clinical phenotype. In a next future, further evolution of ONT chemistry and the optimization of multiplexing strategies are expected to drastically decrease the costs of the analysis, making the Cas9-mediated sequencing more easily accessible in the clinical practice.

Taken together, our pilot study has demonstrated the potential of PCR-free long-read sequencing for the genetic assessment of DM2, allowing us to investigate both the length and genetic features of normal and expanded alleles in a single round of analysis. The use of such an approach in larger

cohorts will increase the accuracy of genotype–phenotype correlations and enhance the information content available for DM2 genetic counseling.

# Materials and methods

**Key resources table**

| Reagent type (species) or resource | Designation | Source or reference | Identifiers | Additional information |
|---|---|---|---|---|
| Gene (*Homo sapiens*) | *CNBP* | Ensembl | HGNC:13164 | Hg38 |
| Biological sample (*Homo sapiens*) | Anti-coagulated peripheral blood | Policlinico Tor Vergata, Rome, Italy | Patient A1, A2, A3, A4, B, C, D, E, F | |
| Sequence-based reagent | Digoxigenin (DIG)-labeled locked nucleic acid (LNA) probe | *Nakamori et al., 2009* | DIG-LNA probe | (CCTG)$_5$ |
| Sequence-based reagent | P4TCTG | This paper | PCR primers | agc gga taa caa ttt cac aca gga TCT GTC TGT CTG TCT GTC TGT |
| Sequence-based reagent | CL3N58_DR-[FAM] | This paper | PCR primers | GCC TAG GGG ACA AAG TGA GA |
| Sequence-based reagent | P3 | This paper | PCR primers | AGC GGA TAA CAA TTT CAC ACA GGA |
| Sequence-based reagent | crRNA1_CNBP | This paper | CRISPR RNA | CCA CCT GAT TCA CTG CGA TA |
| Sequence-based reagent | crRNA2_CNBP | This paper | CRISPR RNA | GGC TTC TCA TTC CAC GAC CA |
| Sequence-based reagent | Native barcodes | Oxford Nanopore Technologies (ONT) | EXP-NBD104 | |
| Commercial assay or kit | DIG-High Prime DNA Labeling and Detection Starter Kit II | Roche | Cat. No. 11585614910 | |
| Commercial assay or kit | Flexigene DNA Kit | Qiagen | Cat. No. 51206 | |
| Commercial assay or kit | BigDye Terminator v3.1 Cycle Sequencing Kit | Thermo Fisher | Cat. No. 4337458 | |
| Commercial assay or kit | Nanobind CBB Big DNA HMW Kit | Circulomics | SKU 102-301-900 | |
| Commercial assay or kit | NucleoSpin Blood L Kit | Macherey-Nagel | Item number: 740954.20 | |
| Commercial assay or kit | Qubit dsDNA BR Assay Kit | Thermo Fisher Scientific | Cat. No. Q32853 | |
| Commercial assay or kit | TapeStation DNA ScreenTape & Reagents | Agilent Technologies | Cat. No. 5067–5365 5067–5366 | |
| Software, algorithm | GeneMapper Software 6 | Applied Biosystems | Cat. No. 4475074 | |
| Software, algorithm | CHOPCHOP | *Labun et al., 2019* | | chopchop.cbu.uib.no/ |
| Software, algorithm | Guppy v3.4.5 | Computational Biology Research Center – AIST | | |
| Software, algorithm | NanoFilt v2.7.1 | *De Coster et al., 2018* | | |
| Software, algorithm | BBMap suite v38.87 | https://sourceforge.net/projects/bbmap/ | | |
| Software, algorithm | Tandem Repeat Finder v4.09 | *Benson, 1999* | | |
| Software, algorithm | Minimap2 v2.17-r941 | Li, 2018 | | |
| Software, algorithm | Integrative Genomics Viewer (IGV) v2.8.3 | Robinson, 2011 | | |

*Continued on next page*

*Continued*

| Reagent type (species) or resource | Designation | Source or reference | Identifiers | Additional information |
|---|---|---|---|---|
| Software, algorithm | Scripts for the generation of consensus sequences and repeat annotations for the normal allele | https://github.com/MaestSi/CharONT2 | | Script Name: CharONT2 |
| Software, algorithm | Scripts for the annotation of repeats and the generation of simplified reads for the expanded allele | https://github.com/MaestSi/MosaicViewer_CNBP | | Script Name: MosaicViewer_CNBP v1.0.0 |
| Software, algorithm | MinKNOW V20.06.5 | Oxford Nanopore Technologies | | |
| Commercial assay or kit | Alt-R S.p. HiFi Cas9 Nuclease v3 | IDT | Cat. No. 1081060 | Recombinant Cas9 nuclease for target excision (see M&M) |
| Commercial assay or kit | Alt-R CRISPR-Cas9 tracrRNA | IDT | Cat. No. 1072532 | Structural RNA for gRNA formation (see M&M) |
| Commercial assay or kit | AMPure XP Beads | Beckman-Coulter | Product No. A63881 | Magnetic beads for nucleic acid purification (see M&M) |
| Commercial assay or kit | CutSmart buffer 10× | New England BioLabs | Cat. No. B7204 | Buffer for gDNA dephosphorylation, RNP formation, and target excision (see M&M) |
| Commercial assay or kit | Blunt/TA Ligase Master Mix | New England BioLabs | Cat. No.: M0367S | T4 DNA ligase for native barcode ligation to dA-tailed ends (see M&M) |
| Commercial assay or kit | FLO-MIN106D (R9.4.1) flow cell | Oxford Nanopore Technologies | FLO-MIN106D | Flowcell for ONT sequencing (see M&M) |

## DM2 patients

We retrospectively analyzed nine genetically confirmed DM2 Italian patients, whose enrollment in the study was approved by the institutional review board of Policlinico Tor Vergata (document no. 232/19). All experimental procedures were carried out according to The Code of Ethics of the World Medical Association (Declaration of Helsinki). Informed consent was obtained from all nine participants and all samples and clinical information were anonymized immediately after collection using a unique alphanumeric identification code. Sociodemographic data for the DM2 patients are summarized in *Table 1*.

## Southern blotting

Genomic DNA was extracted from 500 µl of anticoagulated peripheral blood using a Flexigene DNA Kit (Qiagen, Hilden, Germany) and diluted to a final volume of 25 µl with double-distilled water. The quality and quantity of DNA were assessed using a Denovix spectrophotometer and by 1% agarose gel electrophoresis. *CNBP* expanded alleles were detected as previously described (*Nakamori et al., 2009*), with modifications. Briefly, 2 µg of genomic DNA was digested with AluI and HaeIII and the fragments were resolved by 0.4% agarose gel electrophoresis at 40 V for 40 hr. After denaturation and neutralization, the DNA was transferred to a nylon membrane (MilliporeSigma, Burlington, MA) and fixed by UV cross-linking using a Stratalinker 2400 (Stratagene, San Diego, CA). The membrane was hybridized for 16 hr at 65°C with a digoxigenin (DIG)-labeled locked nucleic acid (LNA) probe $(CCTG)_5$ at a concentration of 10 pmol/ml. After washing at high stringency, the signal was revealed using the DIG High Prime DNA Labeling and Detection Starter Kit II (Roche, Basel, Switzerland) and visualized using an ImageQuant LAS 4000 device (GE Healthcare, Chicago, IL). Bands were sized by running two sets of molecular weight markers alongside the samples: DNA Molecular Weight Marker XV (Expand DNA Molecular Weight Marker, Roche) and $\lambda$ DNA-HindIII Digest (New England Biolabs, Ipswich, MA).

## SR-PCR, QP-PCR, and Sanger sequencing

SR-PCR products were generated as reported earlier (*Kamsteeg et al., 2012*; *Botta et al., 2006*). QP-PCR targeting the 3′ end of the $(CCTG)_n$ repeat array was carried out as previously described (*Catalli et al., 2010*; *Musova et al., 2009*), with modifications. Specifically, the repeat primer P4TCTG-agc gga taa caa ttt cac aca gga TCT GTC TGT CTG TCT GTC TGT (lower case letters indicate the primer

tail that does not complement the repeat) was combined with primers CL3N58_DR-[FAM]-GCC TAG GGG ACA AAG TGA GA and P3-AGC GGA TAA CAA TTT CAC ACA GGA to target the most 3' (TCTG)$_n$ interruptions. The length of the *CNBP* unexpanded alleles and QP-PCR products were determined by capillary electrophoresis on the 3500 Genetic Analyzer followed by analysis using GeneMapper 6 (Applied Biosystems, Waltham, MA). The SR-PCR and QP-PCR products were purified using the ExoSAP protocol, directly sequenced using the Big Dye Terminator Cycle Sequencing Kit v3.1 (Thermo Fisher Scientific, Waltham, MA) and visualized by capillary electrophoresis on the 3500 Genetic Analyzer as above.

## Cas9-mediated enrichment coupled to ONT sequencing

For DM2 patients A1–A4, B, and C, genomic DNA was extracted from 0.2 to 0.5 ml of peripheral blood using the Nanobind CBB Big DNA HMW Kit (Circulomics, Baltimore, MD), designed for HMW DNA extraction. For DM2 patients D, E, and F, the Nanobind CBB Big DNA HMW Kit failed, likely due to the presence of partially degraded DNA consequent to long-term blood storage. For DM2 patients D, E, and F genomic DNA was thus extracted from 1 to 2 ml whole blood using the NucleoSpin Blood L Kit (Macherey-Nagel, Düren, Germany), an extraction kit providing higher yield thanks to the capability of retaining both long and short DNA molecules. Regardless of the extraction method, the DNA was resuspended in Tris–EDTA buffer (pH 8.0) and the quantity was determined using a Qubit fluorometer (Thermo Fisher Scientific) and Qubit dsDNA BR Assay Kit (Thermo Fisher Scientific). DNA integrity was assessed using a TapeStation 4150 device, Genomic DNA ScreenTape and Genomic DNA Reagents (ladder and sample buffer) all from Agilent Technologies (Santa Clara, CA).

We designed crRNAs using the online tool CHOPCHOP (https://chopchop.cbu.uib.no/) following ONT's recommendations (https://community.nanoporetech.com/info_sheets/targeted-amplification-free-dna-sequencing-using-crispr-cas/v/eci_s1014_v1_reve_11dec2018), and making sure that the excised fragment was at least 3 kbp in length. Candidate crRNAs were manually checked for unique mapping by aligning them to the human genome (Hg38) using BLAST and excluding regions overlapping common single-nucleotide polymorphisms (MAF >0.01, dbSNP database). The final crRNAs were prepared by Integrated DNA Technologies (Coralville, IA): 5'-CCA CCT GAT TCA CTG CGA TA-3' with genomic coordinates Chr3:129,175,929–129,175,948, and 5'-GGC TTC TCA TTC CAC GAC CA-3' with genomic coordinates Chr3:129,171,664–129,171,683. The reaction mixture comprised 10 µM of each crRNA, 10 µM of transactivation crRNA (tracrRNA) and 62 µM Alt-R S.p. HiFi Cas9 Nuclease v3 in 1× CutSmart Buffer for the generation of ribonucleoprotein (RNP) complexes (all components from Integrated DNA Technologies) according to the ONT protocol (version ENR_9084_v109_revD_04Dec2018).

The dephosphorylation, Cas9-mediated digestion, and dA-tailing of 1–10 µg input genomic DNA were carried out according to the ONT protocol. The genomic DNA was incubated with the RNPs for 20 min at 37°C and then 2 min at 80°C for enzyme inactivation. ONT sequencing adapters (AMX) were ligated to the cleaved and dA-tailed target ends for 10 min at room temperature before stopping the reaction by adding one volume of 10 mM Tris–EDTA (pH 8.0). Short fragments (<3 kbp) and residual enzymes were removed by adding 0.3× AMPure XP beads (Beckman-Coulter, Brea, CA) and washing twice in long fragment buffer (ONT). The DNA was eluted by incubating for 10 min at room temperature in elution buffer (ONT). Cas9-multiplexing experiments were carried out as described above but EXP-NBD104 native barcodes (ONT) were ligated to the cleaved and dA-tailed target ends using Blunt/TA Ligase Master Mix (New England Biolabs). Samples were quantified and all available nanograms were pooled in a final volume of 65 µl nuclease-free water. ONT sequencing adapters (AMXII from EXP-NBD104) were ligated, followed by purification, washing, and elution as described above for the singleplex experiments. The purified library was mixed with 37.5 µl sequencing buffer (ONT) and 25.5 µl of library loading beads (ONT). The library was loaded onto a FLO-MIN106D (R9.4.1) flow cell and sequenced using MinKNOW v20.06.5 (ONT) until a plateau was reached.

## ONT sequence data analysis

Raw fast5 files were base called using Guppy v3.4.5 in high-accuracy mode, with parameters '-r -i $FAST5_DIR -s $BASECALLING_DIR --flowcell FLO-MIN106 --kit SQK-LSK109 --pt_scaling TRUE' (the last parameter was recommended by ONT technical support to achieve the most appropriate scaling of reads with biased sequence composition, as expected for repeat motifs with low complexity). Reads from multiplexed runs were demultiplexed using Guppy v3.4.5 with parameters

'-i $BASECALLING_DIR -s $DEMulTIPLEXING_DIR `--trim_barcodes --barcode_kits` EXP-NBD104'. Quality filtering was carried out using NanoFilt v2.7.1 (*De Coster et al., 2018*), requiring a minimum quality score of 7. Reads spanning the full repeat were identified using a combination of the scripts msa.sh and cutprimers.sh from BBMap suite v38.87 (https://sourceforge.net/projects/bbmap/). In particular, we used in silico PCR with 100 bp primers annealing to the microsatellite flanking regions (minimum alignment identity = 80%), at least 600 bp up-/downstream, to extract only those reads containing the repeat entirely (https://github.com/MaestSi/MosaicViewer_CNBP/blob/main/Figures/CNBP_left_right_alignment.png; *Maestri, 2020*). These 'complete sequences' were extracted, aligned to either the normal or expanded allele based on length, and subsequent analysis was carried out using a *k*-means clustering method (*k* = 2). Reads from each allele and each sample were then processed separately. Length variability of normal and expanded alleles was determined using the coefficient of variation, measured by the ratio of the standard deviations to the average of the normal/expanded allele lengths.

An accurate consensus sequence was generated by collapsing reads from the normal allele as previously described, using the medaka 'r941_min_high_g344' model (*Grosso et al., 2021*). Finally, the polished consensus sequence was screened for repeats using Tandem Repeat Finder v4.09 (*Benson, 1999*). Scripts for the generation of consensus sequences and repeat annotations for the normal allele have been deposited online (https://github.com/MaestSi/CharONT2; *Maestri, 2022a*).

Reads from the expanded allele were aligned to sequences flanking the repeat, screened for repeats containing the motifs 'TG', 'CCTG', and/or 'TCTG', and visualized in a genome browser (*Grosso et al., 2021*). The extracted sequences and repeat summary file were then imported into R and used to generate a simplified version of the reads with a custom script. In particular, read coordinates corresponding to annotated repeats were replaced with a single-nucleotide stretch equal in length to the annotated repeat, with each motif corresponding to a different nucleotide stretch. In more detail, the flanking region was reported as unchanged to simplify the alignment of reads, 'TG' was replaced with 'GG', 'CCTG' with 'CCCC', 'TCTG' with 'TTTT', and remaining nonannotated portions were replaced with stretches of 'N' of the same length. Reverse complements were generated for simplified reads originally in the reverse orientation, and all simplified reads were aligned to the flanking sequence using Minimap2 v2.17-r941 and visualized in Integrative Genomics Viewer (IGV) v2.8.3. Scripts for the annotation of repeats and the generation of simplified reads for the expanded allele have been deposited online (https://github.com/MaestSi/MosaicViewer_CNBP; *Maestri, 2022b*).

## Acknowledgements

This research was funded by the Muscular Dystrophy Association (https://www.mda.org/). We also thank the Italian DiMio onlus association of DM patients (https://www.dimio.it/) for funding to support research projects.

## Additional information

### Competing interests

Massimo Delledonne: is a partner of Genartis srl. Marzia Rossato: is a partner of Genartis srl, Verona. The other authors declare that no competing interests exist.

### Funding

| Funder | Grant reference number | Author |
|---|---|---|
| Muscular Dystrophy Association | MDA 876149 | Massimo Delledonne |
| Italian DiMio onlus association of DM patients (www.dimio.it) | volunteer donation | Annalisa Botta |

The funders had no role in study design, data collection, and interpretation, or the decision to submit the work for publication.

## Author contributions
Massimiliano Alfano, Investigation, Methodology, Writing - original draft, Writing - review and editing; Luca De Antoni, Investigation, Writing - original draft; Federica Centofanti, Virginia Veronica Visconti, Validation; Simone Maestri, Software, Methodology, Writing - original draft; Chiara Degli Esposti, Maria Rosaria D'Apice, Formal analysis; Roberto Massa, Resources; Giuseppe Novelli, Supervision; Massimo Delledonne, Conceptualization, Supervision, Funding acquisition, Methodology; Annalisa Botta, Conceptualization, Supervision, Funding acquisition, Writing - review and editing; Marzia Rossato, Conceptualization, Supervision, Funding acquisition, Methodology, Writing - original draft, Writing - review and editing

## Author ORCIDs
Simone Maestri http://orcid.org/0000-0002-1192-0684
Chiara Degli Esposti http://orcid.org/0000-0002-5974-4915
Giuseppe Novelli http://orcid.org/0000-0002-7781-602X
Annalisa Botta http://orcid.org/0000-0003-4031-5624
Marzia Rossato http://orcid.org/0000-0002-6101-1550

## Ethics
The study was approved by the institutional review board of Policlinico Tor Vergata (document no. 232/19). All experimental procedures were carried out according to The Code of Ethics of the World Medical Association (Declaration of Helsinki). Informed consent was obtained from all nine participants and all samples and clinical information were anonymized immediately after collection using a unique alphanumeric identification code.

## Decision letter and Author response
Decision letter https://doi.org/10.7554/eLife.80229.sa1
Author response https://doi.org/10.7554/eLife.80229.sa2

# Additional files

## Supplementary files
• Supplementary file 1. Sequencing statistics of singleplex and multiplex experiments. The table reports the feature of each Cas9-mediated sequencing experiment performed and the sequencing statistics of each ONT run. Average values are also provided for singleplex and multiplex runs separately.

• MDAR checklist

## Data availability
The sequencing data generated in this study have been submitted to the NCBI BioProject database under accession number PRJNA818354 (https://www.ncbi.nlm.nih.gov/bioproject/PRJNA818354).

The following dataset was generated:

| Author(s) | Year | Dataset title | Dataset URL | Database and Identifier |
|---|---|---|---|---|
| Alfano M | 2022 | Characterization of full-length CNBP expanded alleles in myotonic dystrophy type 2 patients by Cas9-mediated enrichment and nanopore sequencing | https://www.ncbi.nlm.nih.gov/bioproject/PRJNA818354 | NCBI BioProject, PRJNA818354 |

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
