## [Editor Report]

The study utilized a long-read sequencing to correctly assess expanded alleles in DM2 patients and has technical and diagnostic value. We anticipate more valuable approaches like this in the near future.

---

## [Decision Letter]

**Decision letter after peer review:**

Thank you for submitting your article "Characterization of full-length *CNBP* expanded alleles in myotonic dystrophy type 2 patients by Cas9-mediated enrichment and nanopore sequencing" for consideration by *eLife*. Your article has been reviewed by 2 peer reviewers, including Murim Choi as Reviewing Editor and Reviewer #1, and the evaluation has been overseen by Martin Pollak as the Senior Editor. The following individual involved in the review of your submission has agreed to reveal their identity: Jangsup Moon (Reviewer #2).

Essential revisions:

The reviewers think that this is an interesting study in which the authors conducted Cas9-mediated enrichment and nanopore sequencing to characterize the full-length CNBP expanded alleles in nine myotonic dystrophy type 2 (DM2) patients. However, several improvements are required to warrant publication in *eLife*. The reviewers were concerned with the following issues:

(1) Novelty and clinical utility of the study needs to be emphasized.

(2) More details on technical aspects.

*Reviewer #1 (Recommendations for the authors):*

Clear visualization of repeat status and how it is altered in each patient would help readers understand the results, including a pedigree of the familial cases.

It would help to assess the practical advantages of their method: is it cheaper and faster than the conventional method?

*Reviewer #2 (Recommendations for the authors):*

Overall, this study was well performed and the manuscript was well written, however, there are several points that need to be clarified by the authors.

1. Regarding the (TCTG)n repeats at the 3'end

The percentage of reads carrying the TCTG motif is demonstrated in Table 2, but the fact that the existence of mosaicism can be assumed from this data is not sufficiently explained in the result section. Please include some explanation about the mosaicism.

Is there any possibility that the result shown in Figure 3 is caused by damaged (fragmented) DNA rather than mosaicism? Please explain.

2. In the method section, regarding Cas9 sequencing

Is there any reason why two different DNA extraction methods have been used in patients?

Is there any basis for determining which method to apply to each patient? Please explain.

---

## [Author Response]

Essential revisions:The reviewers think that this is an interesting study in which the authors conducted Cas9-mediated enrichment and nanopore sequencing to characterize the full-length CNBP expanded alleles in nine myotonic dystrophy type 2 (DM2) patients. However, several improvements are required to warrant publication in eLife. The reviewers were concerned with the following issues:(1) Novelty and clinical utility of the study needs to be emphasized.

The novelty and clinical utility of the study have been now better emphasized in the Discussion section. Please refer for more details to replies to point 1, 2 and 6 given to reviewer 1.

(2) More details on technical aspects.

More details on technical aspects have been now provided both in the Results and in the Material and Method section. Please refer for more details to replies to point 3 and 5 given to reviewer 1 and to replies to point 1 and 2 given to reviewer 2.

Reviewer #1 (Recommendations for the authors):Clear visualization of repeat status and how it is altered in each patient would help readers understand the results, including a pedigree of the familial cases.

The repeat status identified by Cas9-mediated sequencing has been now reported more clearly in Table 3. For clarity, the altered quadruplet motifs of each patient have been highlighted with different colours. For the familial cases, we have also included additional figures showing the pedigree (Figure 1—figure supplement 1) and a magnification of reads aligned at the microsatellite locus (Figure 3—figure supplement 2), to show that all family members carry the same motif structure at the repeat 5’-end.

It would help to assess the practical advantages of their method: is it cheaper and faster than the conventional method?

Although potentially applicable for the clinical diagnostics, to date the costs of Cas9-mediated sequencing for a single patient are relatively high and not comparable with the PCR-based approaches used in the routine of the DM2 molecular diagnostics. Nevertheless, the method represents a step forward in DM2 diagnosis, because it will allow to unravel those molecular aspects invisible to conventional methods and thus complement them. In particular, it will help to solve unusual large and complex (CCTG)n expansions not detectable with conventional methods and to identify non-canonical repetitive motifs and sequence interruptions, not visible with other approaches. Taken together, this information will allow more precise correlation between the length and composition of DM2 expansion and the clinical phenotype. In a next future, further evolution of ONT chemistry and the optimization of multiplexing strategies are expected to drastically decrease the cost for the analysis, making the Cas9-mediated sequencing more easily accessible in clinical practice. In terms of turn-around-time, this is already compatible with the diagnostic settings and comparable to conventional methods (1-2 days).

Considerations related to the clinical utility of the approach have been now included in the “Discussion” section (lines 301-307 and lines 427-435).

Reviewer #2 (Recommendations for the authors):Overall, this study was well performed and the manuscript was well written, however, there are several points that need to be clarified by the authors.1. Regarding the (TCTG)n repeats at the 3'endThe percentage of reads carrying the TCTG motif is demonstrated in Table 2, but the fact that the existence of mosaicism can be assumed from this data is not sufficiently explained in the result section. Please include some explanation about the mosaicism.Is there any possibility that the result shown in Figure 3 is caused by damaged (fragmented) DNA rather than mosaicism? Please explain.

We recognize that the result section describing the observed mosaicism was too concise. To address this reviewer’s concern, we have now included a more detailed description and provided an example helping the interpretation of results shown in Table 3 and Figure 3 and Figure 3—figure supplement 1 (see Results, Lines 215-219). For the analysis of expanded microsatellites, we considered only those reads containing at least 600bp up/down-stream the *CNBP* microsatellite, and thus comprising the repeat entirely; we therefore exclude that the extreme variability in molecule length could be ascribed to the analysis of fragmented DNA molecules. This has been now better specified both in the Results (Lines 217-219) and in the Material and Method section (Lines 534-538).

2. In the method section, regarding Cas9 sequencingIs there any reason why two different DNA extraction methods have been used in patients?Is there any basis for determining which method to apply to each patient? Please explain.

Available blood samples of DM2 patients derived from a retrospective cohort and it was stored in our biobank since long time (years). Moreover, the collection and storage of these samples were not conducted with the required precautions for HMW DNA extraction, as preferable for ONT sequencing. Therefore, we could obtain DNA with a kit designed for HMW extraction (Nanobind CBB Big DNA HMW Kit, Circulomics) only from a fraction of patients. When this kit failed, likely due to the presence of partially degraded DNA, we succeeded in the extraction with a more traditional extraction kit (NucleoSpin Blood L Kit, Macherey-Nagel), also yielding long DNA molecules but retaining even shorter ones and thus facilitating the recovery. This aspect has been now clarified in the Material and Method section (lines 483-489).

References

Day JW, Ricker K, Jacobsen JF, Rasmussen LJ, Dick KA, Kress W, Schneider C, Koch MC, Beilman GJ, Harrison AR, Dalton JC, Ranum LP. Myotonic dystrophy type 2: molecular, diagnostic and clinical spectrum. Neurology. 2003 Feb 25;60(4):657-64. doi: 10.1212/01.wnl.0000054481.84978.f9. PMID: 12601109.

Mitsuhashi S, Frith MC, Matsumoto N. Genome-wide survey of tandem repeats by nanopore sequencing shows that disease-associated repeats are more polymorphic in the general population. BMC Med Genomics. 2021 Jan 7;14(1):17. doi: 10.1186/s12920-020-00853-3. PMID: 33413375; PMCID: PMC7791882.

Shruti V Iyer, Melissa Kramer, Sara Goodwin, W. Richard McCombie. ACME: an Affinity-based Cas9 Mediated Enrichment method for targeted nanopore sequencing. bioRxiv 2022.02.03.478550; doi: https://doi.org/10.1101/2022.02.03.478550

Stevanovski I, Chintalaphani SR, Gamaarachchi H, Ferguson JM, Pineda SS, Scriba CK, Tchan M, Fung V, Ng K, Cortese A, Houlden H, Dobson-Stone C, Fitzpatrick L, Halliday G, Ravenscroft G, Davis MR, Laing NG, Fellner A, Kennerson M, Kumar KR, Deveson IW. Comprehensive genetic diagnosis of tandem repeat expansion disorders with programmable targeted nanopore sequencing. Sci Adv. 2022 Mar 4;8(9):eabm5386. doi: 10.1126/sciadv.abm5386. Epub 2022 Mar 4. PMID: 35245110; PMCID: PMC8896783.

Vanacore N, Rastelli E, Antonini G, Bianchi ML, Botta A, Bucci E, Casali C, Costanzi-Porrini S, Giacanelli M, Gibellini M, Modoni A, Novelli G, Pennisi EM, Petrucci A, Piantadosi C, Silvestri G, Terracciano C, Massa R. An Age-Standardized Prevalence Estimate and a Sex and Age Distribution of Myotonic Dystrophy Types 1 and 2 in the Rome Province, Italy. Neuroepidemiology. 2016;46(3):191-7. doi: 10.1159/000444018. Epub 2016 Feb 17. PMID: 26882032.